# Chronic Diseases and Influenza Vaccines

**DOI:** 10.3390/vaccines13090936

**Published:** 2025-09-01

**Authors:** Rui Lian, Hongbo Zhang, Youcai An, Ze Chen

**Affiliations:** 1Emergency Department of China-Japan Friendship Hospital, Beijing 100029, China; lianrui@cjfh.org.cn; 2Department of Basic Research, Ab &B Bio-Tech Co., Ltd. JS, Taizhou 225300, China; zhanghongbo@abbbio.com.cn (H.Z.); anyoucai@abbbio.com.cn (Y.A.); 3Innovative Human Vaccine Technology and Engineering Research Center of Taizhou, Taizhou 225300, China; 4Innovative Antiviral Vaccines Engineering Technology Research Center of Taizhou, Taizhou 225300, China

**Keywords:** chronic diseases, influenza, immune dysregulation, vaccination, complications

## Abstract

Background: Chronic illnesses pose a major global health challenge with an estimated 1.56 billion people affected worldwide in 2025, and 85% of these being older adults facing at least one chronic condition. These patients are particularly vulnerable to severe influenza complications and higher mortality rates due to weakened immune responses; in addition, vaccination rates in China remain significantly lower than those in developed nations. Methods: This review examines how chronic conditions exacerbate influenza-related effects through immune dysfunction and metabolic imbalances, and how influenza infection worsens chronic diseases by triggering inflammation, suppressing immunity, and causing secondary infections that lead to respiratory complications, cardiac complications, and blood sugar disturbances. Results: A bidirectional adverse interaction exists in which chronic illnesses increase influenza severity via poor immunity, while influenza accelerates chronic disease progression (e.g., cardiac events and diabetic ketoacidosis). Vaccination reduces hospitalization by 32–52% in patients with lung disease and mortality by 16–46% in diabetic patients, with good safety. Conclusions: The findings emphasize the urgent need for improved vaccination strategies in patients with chronic diseases. Such strategies are crucial to reducing disease burden, enhancing clinical outcomes, and improving quality of life, while also providing critical evidence for the development of public health policies.

## 1. Introduction

Chronic diseases, also known as chronic non-communicable diseases (NCDs), refer to a general term for diseases characterized by insidious onset, a prolonged course, and lack of definitive evidence of infectious biological causes [1]. Chronic diseases mainly include cardiovascular and cerebrovascular diseases, diabetes, cancer, and chronic respiratory diseases [2].

By 2025, the global prevalence of chronic diseases is expected to reach 1.56 billion [3,4]. A key contributor to this trend is the aging of the population, with the global proportion of people aged 65 and older expected to increase to 16% by 2050 [5]. Among this aging demographic, an estimated 85% will suffer from at least one chronic disease, and 60% will be diagnosed with two or more [6].

China is facing particularly challenging situation regarding chronic disease prevention and control, characterized by three key features: firstly, the overall prevalence rate continues to escalate, with an estimated 500–600 million individuals requiring health management; secondly, population aging exacerbates the problem, as over 50% of those aged 60 and above are afflicted, accounting for 130 million patients and approximately 21.7% of the total chronic disease burden; thirdly, chronic non-communicable diseases (NCDs) such as cardiovascular and cerebrovascular diseases and malignancies have emerged as major public health threats [7]. Patients with chronic diseases require long-term treatment, care, and rehabilitation, imposing huge pressure on families in terms of medical expenses [8]. Many families, especially those in rural areas, fall into poverty due to chronic diseases [9]. Nationally, chronic diseases account for 70% of total healthcare expenditures, creating a significant strain on the public healthcare system [10]. In addition, indirect economic losses such as labor loss and reduced ability to work caused by chronic diseases are also significant, affecting social and economic development [11].

Influenza is acute, highly contagious, and rapidly spreading, and it is one of the main causes of human death [12]. Among the three types of influenza viruses (A, B, and C), influenza A exhibits the broadest host range and poses the greatest public health threat [13]. This virus is responsible for seasonal epidemics and has the potential to trigger global pandemics. Since the 20th century, there have been five global influenza pandemics [14]. Notably, the 2009 H1N1 influenza A (H1N1) virus triggered the first influenza pandemic of the 21st century, affecting 214 countries and regions and causing at least 18,449 deaths [15,16]. Each influenza pandemic has brought catastrophic impacts to human life, property, and economic development [17,18]. Due to the high transmissibility and rapid mutation rate of influenza viruses, humans have not been able to effectively prevent influenza viruses thus far. Consequently, annual influenza vaccination remains the only effective measure to prevent seasonal influenza [19,20,21].

It is particularly noteworthy that patients with chronic diseases are prone to severe complications or even death after influenza virus infection due to compromised immune function [22,23,24]. According to statistics from the World Health Organization (WHO), approximately 60% of global influenza-related fatalities annually occur in patients with chronic diseases [25,26]. Influenza vaccination markedly reduces the risk of hospitalization and mortality connected to influenza and its related complications in patients with chronic conditions [27]. However, the influenza vaccination rate among patients with chronic disease in China is far lower than that in developed countries [28]. Among patients with chronic diseases, the influenza vaccination rate is approximately 11.5% in China, whereas it ranges from 50% to 60% in developed countries. Therefore, in-depth exploration of the necessity and optimized strategies for influenza vaccination in patients with chronic disease is of great importance for public health [29].

## 2. Global and China’s Current Chronic Disease Burden

By 2025, it is estimated that 1.56 billion people worldwide will have hypertension, and three-quarters of them will be in developing countries [30]. In Asia, the number of people with type 2 diabetes is increasing by 3.5% every year [31]. Cardiovascular and cerebrovascular diseases affect 15% to 20% of the population, and they are becoming more common among younger people [32,33,34]. In total, 18% of patients are under 40 years old. By 2030, the number of patients will reach 774 million, and the number of deaths will rise to 22.2 million, with more than half due to strokes and 45% due to coronary heart disease [35,36]. The number of new cancer cases worldwide is expected to increase from 18 million in 2018 to 27 million in 2040, reaching 35.281 million in 2050, which is an increase of 76.6% compared to 2022, with lung cancer and four other types accounting for 60.8% of new cases [37,38]. Chronic obstructive pulmonary disease affects 210 million people and is the third leading cause of death globally, causing more than 3 million deaths annually [39,40]; the incidence of hyperlipidemia in people over 60 is 39.5% [41]. The overall prevalence of osteoporosis is 18.3%, with 23.1% in women and 11.7% in men [42,43,44]. The prevalence of chronic kidney disease in people over 60 ranges from 10% to 16% [45,46].

In China, the prevalence of various chronic diseases remains high [7,47]. Specifically, hypertension affects over 240 million people, ranking first among chronic diseases. The prevalence rate of diabetes is 20.1%, with more than 230 million patients, and the incidence of diabetes among adolescents and young people has increased significantly [48]. Cardiovascular and cerebrovascular diseases claim 5.085 million lives annually, accounting for 47.8% of total chronic disease-related deaths [49]. Cancer produces over 4 million new cases and 2,818,000 deaths per year [50,51]. For chronic respiratory diseases, primarily chronic obstructive pulmonary disease (COPD), the prevalence rate among individuals aged over 40 has increased by 3.7 percentage points compared to 2015 [52]. Additionally, approximately 160 million people suffer from hyperlipidemia, osteoporosis affects 36% of those aged 60 and above, and the prevalence of chronic kidney disease is around 10.8% [53,54,55,56].

## 3. Influenza Virus Infection and Immune Dysfunction in Chronic Diseases

### 3.1. Immune Dysregulation Induced by Chronic Diseases

Patients with chronic disease frequently exhibit immune dysregulation affecting both humoral and cellular immune responses, typically characterized by elevated proinflammatory cytokine levels [57,58] (Figure 1). Cellular immune abnormalities manifest as T-cell subset dysregulation, particularly the imbalance between Th17 and Treg cells [59,60]. Th17 cells secrete proinflammatory cytokines to mediate chronic inflammatory responses, while the reduction in Treg cell count or functional impairment weakens immune tolerance, leading to persistent inflammation [59,61,62]. Additionally, natural killer (NK) cells show altered immunophenotypes and impaired cytotoxicity, compromising the body’s immune surveillance capacity [63,64,65,66,67,68]. In COPD, CD8^+^ T cells demonstrate reduced mitochondrial membrane potential and glycolytic activity, resulting in diminished cytotoxic capabilities [69,70]. Additionally, an elevated proportion of Th17 cells promotes neutrophil and monocyte infiltration into lung tissues [71]. In autoimmune diseases, the dysregulation of CD4^+^ T-cell subsets plays a critical role [72,73,74]. For example, in rheumatoid arthritis, Th17 cells secrete cytokines such as IL-17, promoting inflammatory responses and causing joint damage [75].

Humoral immune abnormalities primarily reflect changes in immunoglobulin levels, with decreased levels of IgG, IgA, and IgM indicating downregulated immune function [76,77,78,79,80,81]. This not only increases the risk of infection but also disrupts the balance of immune responses, exacerbating the progression of chronic diseases (Table 1).

Significant advances have been made in research on immunological abnormalities in chronic diseases (Figure 1). These abnormalities are particularly evident in immune cell dysfunction, characterized by aberrant T-cell and B-cell differentiation and impaired function. Cytokines serve as critical mediators of immune dysfunction in chronic diseases. Elevated levels of IL-17, IL-6, and TNF-α not only promote inflammatory reactions but also modulate immune cell activation and function, ultimately contributing to the chronicity of disease. Signaling pathway abnormalities further exacerbate these processes, with the aberrant activation of Toll-like receptors (TLRs) and other pattern recognition receptors (PRRs) playing a pivotal role in chronic inflammation and autoimmunity [100,101,104,105,106]. Uncontrolled activation of the TLR family triggers excessive inflammatory responses, driving the development of autoimmune diseases. For instance, TLRs activate the downstream NF-κB signaling pathway, inducing cytokine secretion and enhancing immune responses [107,108]. Sustained activation of the NF-κB pathway is a hallmark of chronic inflammatory and autoimmune diseases, leading to overexpression of proinflammatory cytokines and chemokines that exacerbate inflammatory processes [109,110,111]. In SLE, dysregulated NF-κB activity correlates with disease activity and influences B cell activation and autoantibody production [102,103,112] (Table 1) (Figure 1).

Epigenetic mechanisms also play a crucial role in immunological abnormalities of chronic diseases. Epigenetic alterations such as DNA methylation and histone modifications influence gene expression, leading to dysfunction of immune cells [127,128]. For example, in chronic fatigue syndrome, changes in histone modifications affect the expression of immunity-related genes, thereby impacting immune function [113,114]. Modulating epigenetic states has the potential to improve immunological abnormalities in chronic diseases [129,130]. Certain drugs regulate immune cell function and alleviate chronic inflammation and autoimmune responses by influencing epigenetic modifications [131,132]. Histone deacetylase inhibitors (HDACIs), for instance, regulate gene expression by altering histone acetylation status, thereby improving immune cell activation and function [133,134,135,136]. Additionally, microenvironmental changes in patients with chronic disease significantly impact immune cell activation and function [137,138]. Aberrant expression of cytokines, chemokines, and other signaling molecules in the microenvironment promotes chronic inflammation and autoimmune responses [139]. In COPD, persistent airway inflammation releases inflammatory mediators and chemokines, exacerbating airway remodeling and lung tissue destruction. Intervening in the microenvironment can regulate immune responses and ameliorate disease progression [140,141,142,143]. Anti-inflammatory drugs or immunomodulators can reduce levels of specific cytokines, improve immune cell activation and function, and mitigate disease symptoms and progression [144,145,146,147,148] (Table 1) (Figure 1).

Molecular biological research on immunological abnormalities in chronic diseases has unveiled aberrant changes in immune cells, cytokines, and signaling pathways. Recent advances highlighting the roles of epigenetic regulation and microenvironmental factors have not only deepened our mechanistic understanding but also established a robust theoretical framework for developing innovative therapeutic approaches targeting these pathological processes.

### 3.2. Direct Immunopathological Injury and Hypersensitive Reactions Induced by Influenza

#### 3.2.1. Cytokine Storm

Upon infection with influenza viruses (such as H5N1, H7N9, and 2009 H1N1), excessive activation of the immune system triggers the massive release of proinflammatory cytokines, leading to alveolar epithelial and vascular endothelial injury, pulmonary edema, and respiratory failure [149,150,151,152,153,154]. These mechanisms underlie the primary causes of death in severe influenza cases.

Upon influenza virus infection, the initiation of immune responses begins with the recognition of viral pathogen-associated molecular patterns (PAMPs) by host pattern recognition receptors (PRRs) [155,156,157]. PRRs identify conserved molecular structures of viral nucleic acids, such as single-stranded RNA (ssRNA) and double-stranded RNA (dsRNA), triggering downstream signaling pathways. Iwasaki and Medzhitov and Herold et al. proposed the central role of PRRs in innate immunity, summarizing three key signaling pathways: the endosomal TLR pathway, cytoplasmic RIG-I pathway, and inflammasome pathway [158,159] (Figure 2). (1) Endosomal TLR3 and TLR7 Pathways: These pathways recognize viral RNA within endosomes, activating transcription factors to induce cytokine and interferon (IFN) production. TLR3 identifies viral dsRNA (e.g., influenza replication intermediates), promoting IFN-β gene expression [160]. TLR7 recognizes viral ssRNA, stimulating NF-κB and IRF7 to induce proinflammatory cytokines (e.g., IL-6) and type I interferons (e.g., IFN-α) [161]. (2) Cytoplasmic Retinoic Acid-Inducible Gene-I (RIG-I) Pathway: RIG-I recognizes viral ssRNA in the cytoplasm and initiates antiviral immune responses. The RIG-I protein specifically binds to viral ssRNA with a 5′-triphosphate group, activating IRF3/IRF7 and NF-κB to promote transcription of cytokines (e.g., IL-8), chemokines, and interferon-stimulated genes (ISGs). RIG-I functional deficiency is associated with severe infections [162,163] (Figure 2). (3) Inflammasome Pathway: This pathway recognizes viral components to activate maturation and the release of the proinflammatory cytokine IL-1β. The NLRP3 inflammasome, expressed in dendritic cells (DCs), neutrophils, macrophages, and monocytes, detects influenza viruses. Activated caspase-1 cleaves pro-IL-1β into mature IL-1β, triggering local inflammatory responses to recruit immune cells for infection clearance [164,165] (Figure 2).

In the early stage of influenza virus infection, all initial signaling pathways and immune cells involved in resisting influenza benefit the host by preventing viral replication and release [166,167]. However, in severe infections, excessive activation of the immune system leads to a major release of proinflammatory cytokines (e.g., IL-6, IL-1β, IFN-γ), shifting protective immune responses to harmful ones [168,169]. The determinants underlying this transition from protective to pathogenic immunity remain unclear. It is hypothesized that both pathogen characteristics and host factors may differentially contribute to the establishment of cytokine storms.

Influenza virus gene products (PB2, PB1, PB1-F2, PA, PA-x, HA, NP, NA, M1, M2, and NS1) contribute to viral infection and cytokine storms through multi-dimensional mechanisms (Figure 2) [170]:

PB2 (RNA polymerase subunit): Mutations like E627K enhance viral replication in mammalian cells, increasing the viral load to activate TLRs [171]. It inhibits interferon pathways by binding importin-α/β, activating NLRP3 inflammasome to induce proinflammatory cytokines, and recruiting immune cells via chemokines, forming positive feedback loops that disrupt T-cell function. PB2 mutations correlate with clinical severity and cytokine storm intensity, serving as key virulence genes for H7N9 infectivity [172,173,174].

PB1 generates PB1-F2 via +1 ribosomal frameshifting. Avian influenza-derived full-length PB1-F2 (90–110 amino acids) activates NF-κB through TLR pathways to upregulate proinflammatory factors, promotes macrophage polarization to proinflammatory phenotypes, induces Treg apoptosis, and triggers NLRP3 inflammasome-mediated pyroptosis. It forms inflammatory amplification loops with ROS and ATP, damages vascular endothelium, and impairs MAVS-mediated interferon induction. Virulence is modulated by full-length structure and mutations (e.g., E62D) [175,176,177].

PA produces PA-x via ribosomal frameshifting. PA-x’s C-terminal 61/41-amino acid domain inhibits host protein synthesis via endoribonuclease activity, regulating inflammation and viral replication [178]. Deletion in low-pathogenic viruses enhances pathogenicity, while deletion in highly pathogenic viruses promotes replication and stronger cytokine storms [179]. C-terminal integrity and R195K mutation influence inflammatory regulation [180].

HA (hemagglutinin) mediates viral binding to target cells: human influenza HA binds to α2,6-sialic acid and avian HA binds to α2,3-sialic acid to determine host tropism [181,182]. Endosomal acidification triggers HA conformational changes for membrane fusion, and pH stability affects tropism [183,184]. Newly synthesized HA requires proteolytic cleavage into HA1/HA2 for activation. Highly pathogenic avian influenza HA contains multi-basic cleavage sites recognized by widespread proteases [185,186].

NP (nucleoprotein) stimulates neighboring cells via TLR2/4 and NLRP3 inflammasome, inducing IL-1β/IL-6 to upregulate trypsin and enhance viral infectivity [187]. It binds ZBP-1 to activate the NLRP3 inflammasome, inducing lung epithelial cell death and serving as a key virulence gene for H7N9 [188,189].

NA (neuraminidase) enhances HA-mediated membrane fusion and infectivity by removing sialic acids from viral/host membranes to facilitate virion release [190,191,192,193,194]. It inactivates regulatory CD83 pathways via sialic acid cleavage, releasing excessive cytokines for lung injury. Infected dendritic cells show NA-upregulated proinflammatory factors (e.g., CD83) and downregulated anti-inflammatory factors, targetable via NA stalk length, activity, or vaccines [195,196].

M2 binds the autophagy regulator LC3 to inhibit autophagy and enhance viral stability, activating NLRP3 inflammasome to increase pathogenicity [197].

NS1 (key virulence factor) impairs host antiviral responses through multiple mechanisms, regulating cellular processes to suppress immune responses [198,199].

Host factors, including demographic, clinical, and genetic factors, are linked to susceptibility to severe influenza. Age is a major risk factor, with young adults potentially exhibiting hyperactive immune responses that trigger excessive inflammation and older adults experiencing immunosenescence, characterized by impaired viral clearance due to dysregulated immune responses [200,201,202]. Gender also influences influenza outcomes, particularly during pandemics; for example, the 2009 H1N1 pandemic saw a predominance of young women experiencing severe influenza hospitalizations. However, the mechanism by which sex hormones modulate immune responses remains unclear [203,204]. Obesity is associated with higher rates of influenza morbidity and mortality, as elevated leptin and free fatty acids in obese individuals activate Toll-like receptors (TLRs) on monocytes and lymphocytes, driving excessive production of proinflammatory cytokines (e.g., IL-6, TNF-α) and exacerbating inflammatory responses [205,206]. Genetic factors, such as single-nucleotide polymorphisms (SNPs) in genes encoding pattern recognition receptors (PRRs); signaling molecules; transcription factors; and cytokines/chemokines and their receptors, can disrupt immune regulation and predispose individuals to hyperinflammatory responses during influenza infection; these genetic variants interact with pathogen virulence, viral load, and demographic factors to trigger cytokine storms [207,208]. Influenza-induced cytokine storms commonly involve proinflammatory mediators (TNF-α, IFN-γ, IL-1β, IL-2, IL-6, CXCL8, CCL2, CCL3, CXCL10, G-CSF, FGF, VEGF) and anti-inflammatory factors (TGF-β, IL-10, IL-1RA) [168,209].

#### 3.2.2. Immunosuppression

The innate immune system serves as the body’s first line of defense against pathogens, comprising essential immune cell populations such as macrophages, neutrophils, and dendritic cells. The influenza virus infection can directly impair these key effector cells, thereby compromising the host’s early antiviral defense mechanisms [210,211].

While the suppressive effects of influenza on innate immunity are well-documented, emerging research has uncovered a distinct mechanism by which the virus directly suppresses adaptive immune responses [212,213]. Influenza exhibits selective tropism for activated lymphocytes that respond to both influenza and other respiratory pathogens. This targeted infection is enabled by heightened sialic acid expression on activated lymphocytes, which acts as a molecular receptor for influenza virus entry [214,215]. This targeted infection leads to the functional impairment of these lymphocytes, preventing them from differentiating normally into pathogen-specific antibody-secreting cells (ASCs). Therefore, both in vaccinated populations and in animal models, the number of ASCs against influenza viruses and heterologous respiratory pathogens is significantly reduced due to the targeted infection and damage of these activated lymphocytes that should have differentiated into antibody-secreting cells.

These findings reveal that the influenza virus can directly target and impair activated immune cells, inducing a state of immunosuppression that persists even in previously vaccinated hosts. This immunological impairment not only increases susceptibility to severe disease but also provides a mechanistic explanation for vaccine breakthrough infections during influenza seasons by undermining the very immune cells essential for mediating protective humoral responses [216,217].

The influenza virus employs a sophisticated tripartite strategy to antagonize host immune defenses. First, it inhibits interferon (IFN) production through multiple mechanisms: the NS1 protein binds to RIG-I to prevent its ubiquitination and activation, while the Nsp1 protein blocks TBK1/IRF3 phosphorylation to suppress IFN signaling [218]. Second, it induces the shutdown of host gene expressions by disrupting mRNA processing and nuclear export via NS1; it directly inhibits translation and degrades host mRNAs through Nsp1′s ribosome-binding activity [219,220]. Third, it antagonizes interferon-stimulated genes (ISGs) by inhibiting antiviral effectors such as PKR and the OAS-RNase L pathway through NS1 and suppressing TYK2/STAT2 signaling via Nsp1 [221,222,223]. Additionally, the NS1 protein’s functional regulation is modulated by post-translational modifications, including phosphorylation at specific residues (e.g., S42, T49) and SUMOylation, collectively enabling the virus to comprehensively evade host immune responses [224,225].

#### 3.2.3. Secondary Infection

Secondary infections represent a major complication of influenza, primarily resulting from compromised respiratory mucosal barriers and local immune dysfunction [226]. The influenza virus directly damages alveolar epithelial cells, impairing mucociliary clearance and disrupting mucosal integrity, creating favorable conditions for bacterial colonization and invasion [227,228]. This significantly increases susceptibility to secondary bacterial infections, particularly with pathogens such as Streptococcus pneumoniae and Staphylococcus aureus, which often progress to severe bacterial pneumonia and other complications [229,230].

Virus-induced immunosuppression further exacerbates this vulnerability. Influenza infections cause innate immune cell exhaustion, suppress type I interferon production, intensify oxidative stress and cytokine storm responses, and induce lung tissue damage, all of which act synergistically to enhance susceptibility to secondary pathogens [231]. Notably, co-infection with influenza virus and S. pneumoniae has been shown to deplete germinal center B cells, plasma cells, and T cells in lymphoid tissues, resulting in reduced antibody titers. Additionally, impaired antigen-presenting cell (APC) function diminishes bacterial antigen presentation efficiency, further delaying the activation of protective immune responses [232].

### 3.3. Influenza Infection Exacerbates Chronic Diseases: Pathological Mechanisms

A. Influenza infection significantly exacerbates chronic respiratory conditions, including asthma, COPD, and bronchiectasis [233,234,235,236,237]. Notably, pulmonary influenza infections often trigger acute exacerbations of COPD (AECOPD), leading to critical events that substantially diminish patients’ quality of life and reduce survival rates. A landmark Indian study demonstrated that influenza infection is a major contributor to AECOPD, with research revealing that exacerbations are associated with accelerated declines in lung function and up to 23% annual mortality [238] (Table 2).

Historically, viral infections are considered responsible for only a small proportion of COPD exacerbations. However, contemporary research indicates that viruses may actually account for nearly 50% of AECOPD cases, with the influenza virus detected in 28% of exacerbated COPD patients. Influenza has emerged as a particularly important pathogen in severe COPD exacerbations requiring hospitalization, establishing itself as a major determinant of COPD morbidity and mortality [264,265,266,267] (Table 2).

B. Influenza virus infection imposes multifaceted impacts on the cardiovascular system [195,268,269,270,271,272,273,274,275]. This particularly accelerates pathological progression in patients with chronic heart diseases, imposing additional stress on the cardiovascular system—this is manifested as an increased heart rate, elevated blood pressure, and exacerbation of chronic cardiac conditions such as heart failure and coronary artery disease. The core mechanisms involve (1) direct myocardial injury, in which the influenza virus (especially type A) infects cardiomyocytes, leading to viral replication, cell apoptosis/necrosis, and mitochondrial damage that worsens heart failure symptoms [269,276,277]; (2) uncontrolled systemic inflammation damage to vascular endothelium, which increases permeability, and promotes thrombus formation, thereby elevating risks of myocardial infarction and stroke [278,279,280]; and (3) excessive neuroendocrine activation, where infection-induced fever and hypoxia stimulate sympathetic nerves to release catecholamines, causing tachycardia, increased myocardial oxygen consumption, and hypertension that may precipitate acute heart failure or myocardial ischemia in coronary patients [281,282] (Table 2).

C. Influenza infection induces significant metabolic stress, leading to glycemic dysregulation and insulin resistance that exacerbate chronic metabolic conditions such as diabetes and metabolic syndrome [283,284,285,286]. This metabolic dysfunction stems from infection-induced inflammatory responses, stress hormone secretion, and tissue/organ impairment, which operate through four interconnected mechanisms: (1) proinflammatory cytokine release, where the virus triggers immune activation and massive cytokine secretion that disrupt insulin signaling and glucose uptake; (2) elevated stress hormone secretion, as infection-induced stress promotes adrenaline and cortisol release, which antagonizes insulin action while stimulating hepatic gluconeogenesis and glycogenolysis; (3) tissue–organ metabolic dysfunction, including enhanced hepatic gluconeogenesis, reduced muscular glucose utilization, and increased adipose lipolysis, which release free fatty acids and exacerbate insulin resistance; and (4) immune–metabolic axis dysregulation, characterized by the abnormal crosstalk between immune cells and metabolic cells, which creates a self-reinforcing cycle of inflammation and metabolic imbalance, further disrupting glucose homeostasis. These combined effects demonstrate how influenza infection can significantly worsen metabolic disorders through multiple concurrent pathways (Table 2).

D. Influenza virus infection induces significant immune dysregulation that exacerbates chronic inflammatory conditions such as rheumatoid arthritis and inflammatory bowel disease through a complex inflammatory cascade involving innate immune hyperactivation, adaptive immune dysfunction, cytokine network imbalances, and gut–immune axis interactions [23,287,288,289,290]. Systemic inflammation is further amplified by intestinal barrier disruption and dysbiosis via the gut–joint axis, along with oxidative stress and mitochondrial damage driving tissue fibrosis [291,292]. Clinical evidence confirms that influenza significantly increases disease activity in chronic inflammatory patients, demonstrating the clinical importance of these immunological interactions (Table 2).

Immune dysregulation associated with chronic diseases and immunopathological injuries caused by influenza interacts dynamically, forming a vicious cycle. Chronic diseases (see Section 3.1) create a “primed” inflammatory microenvironment characterized by elevated proinflammatory cytokines and impaired mucosal barriers, which amplifies the immunopathological effects of influenza. This pre-existing state hypersensitizes the immune system to viral triggers, exacerbating cytokine storms and tissue damage during influenza infection.

Conversely, influenza disrupts immune homeostasis in these patients (see Section 3.2): the virus’s suppression of adaptive immunity (such as targeted infection of activated lymphocytes) exacerbates pre-existing humoral immune deficiencies, weakening antiviral defense capabilities and the control of underlying diseases. Additionally, influenza-induced epithelial barrier disruption and secondary infections exploit compromised mucosal immunity in chronic respiratory diseases like COPD, accelerating airway remodeling and fibrosis.

This reciprocal interaction increases the risk of severe influenza complications and exacerbation of underlying chronic diseases in patients with chronic illnesses, highlighting the necessity of optimizing influenza vaccination strategies to break this cycle.

## 4. Epidemiology

Influenza poses a disproportionately high risk to vulnerable populations, necessitating targeted public health strategies and clinical interventions. Patients with chronic disease—including those with cardiovascular diseases, diabetes, chronic obstructive pulmonary disease (COPD), and chronic kidney diseases—experience significantly elevated risks of severe influenza complications, with markedly higher hospitalization and mortality rates compared to the general population [293,294,295]. This vulnerability is particularly pronounced among older adults, who exhibit immunosenescence and experience hospitalization and mortality rates that far exceed other age groups, especially when comorbidities are present [296,297]. Notably, nearly 50% of elderly influenza patients have concurrent cardiovascular diseases and COPD, with in-hospital case fatality rates reaching 2.1% [233,298] (Table 3).

Young children also represent a high-risk group due to their immature immune systems, with influenza hospitalization rates as high as 69%—boys face a 1.6-fold greater hospitalization risk than girls [320,321]. Advanced research has identified several key prognostic factors in severe influenza A cases: age, respiratory failure, aspartate aminotransferase levels, and lymphocyte counts. These all serve as predictors of 28-day mortality, particularly concerning outcomes that occur when lymphocyte counts fall below normal thresholds. Multivariate analysis further reveals that D-dimer levels, the PaO_2_/FiO_2_ ratio, and platelet-to-lymphocyte ratio (PLR) independently predict influenza-induced sepsis, with PLR demonstrating the highest diagnostic sensitivity [322].

In summary, hospitalized influenza patients with chronic conditions, especially older adults and young children, exhibit significantly higher mortality rates than the general population. These findings underscore the critical importance of (1) increasing the coverage of influenza vaccinations in patients with chronic diseases (a key high-risk group) to reduce influenza-related hospitalization, disease severity, and mortality compared to their unvaccinated counterparts and (2) strengthening influenza surveillance systems and implementing early intervention strategies to further improve clinical outcomes in this vulnerable population.

## 5. Application of Influenza Vaccines

Vaccination remains the cornerstone of influenza prevention, with the current global landscape featuring three primary vaccine platforms: egg-based inactivated whole-virus vaccines, split vaccines, and subunit vaccines [323,324]. These are typically formulated as trivalent or quadrivalent preparations combining monovalent antigen stocks from circulating influenza A (H1N1, H3N2) and B strains. The history of influenza vaccination dates back to the 1940s. In 1941, the first formalin-inactivated whole-virus influenza vaccine produced in chicken embryos was approved in the United States, followed by its market launch in 1945. However, due to immature purification techniques at the time, this whole-virus vaccine caused severe adverse reactions. It was not until the 1960s that zonal centrifugation, which is a purification method still in use today, significantly improved the quality of inactivated influenza vaccines by enhancing viral antigen purity. In 1968, Pasteur (now Sanofi) pioneered the development of influenza split vaccines globally. Prepared by disrupting inactivated whole viruses with detergents, split vaccines retain only immunogenic components (HA, NA, partial NP, and M) while removing viral macromolecules and nucleic acids. This higher-purity formulation reduces adverse reactions, demonstrating superior safety and immunogenicity compared to early whole-virus vaccines [325,326,327].

Third-generation subunit vaccines were developed in 1976, further refining split vaccines by excluding internal viral proteins to contain highly purified HA and NA. Clinical studies confirmed their enhanced safety profile relative to previous vaccine types [328]. Novel adjuvants have since been applied to augment vaccine immunogenicity, with MF59 (an oil-in-water emulsion adjuvant) approved in over 20 countries worldwide [329]. Additionally, cell-culture-based vaccines, live attenuated influenza vaccines (LAIVs), and mRNA vaccines have entered the developmental pipeline [330,331,332,333,334,335].

## 6. Efficacy of Influenza Vaccines in Patients with Chronic Diseases

Multiple studies have demonstrated that influenza vaccination significantly reduces hospitalization and mortality risks associated with influenza and related complications in patients with chronic diseases compared to unvaccinated patients with the same chronic diseases [27]. A meta-analysis showed that influenza vaccination reduces all-cause mortality by 46% and influenza/pneumonia-related hospitalization by 45% in elderly patients with diabetes compared to unvaccinated elderly patients with diabetes [336]. For individuals with cardiovascular diseases, influenza vaccines exert protective effects by lowering the risk of adverse cardiovascular events, with a 34% reduction in cardiovascular event risks observed among vaccinated patients compared to unvaccinated individuals with cardiovascular diseases. Additionally, the increased incidence of acute myocardial infarction and mortality during influenza seasons highlights the strong association between cardiovascular diseases and influenza infection [337] (Table 4).

In patients with COPD, influenza vaccination significantly decreases the risk of acute exacerbations [345,383]. One year post-vaccination, COPD patients exhibit notable reductions in COPD Assessment Test (CAT) scores, with improved symptoms including coughing, chest tightness, and dyspnea. Importantly, while influenza vaccines show no significant impact on lung function parameters in stable elderly COPD patients, they effectively reduce exacerbation incidence and hospitalization frequency [384] (Table 4).

For patients with heart failure, influenza vaccination lowers both hospitalization and mortality risks. A meta-analysis incorporating six observational cohort studies linked influenza vaccination to reduced mortality during 1-year and long-term follow-ups [385]. Another study indicated that vaccination significantly decreases readmission rates in patients with cardiovascular disease aged ≥65 years [341]. Diabetes mellitus increases influenza infection risk by 3.63-fold compared to non-diabetic populations, making influenza vaccination a key strategy to reduce all-cause mortality and infection-related hospitalizations [302,386,387]. Vaccinated diabetic patients show a 23% reduction in all-cause hospitalization risk and a 45% reduction in influenza/pneumonia-related hospitalization risk [353,385,388] (Table 4).

Influenza vaccines reduce hospitalization and mortality in patients with chronic disease by targeting their unique vulnerabilities, countering immune dysregulation, metabolic imbalances, and organ-specific weaknesses to mitigate influenza-induced exacerbations. Vaccines address inherent immune deficits as follows: in patients with diabetes or cardiac problems with impaired CD8^+^ T-cell activity and excess proinflammatory cytokines (see Section 3.1), they restore CD8^+^ T-cell-mediated clearance and induce virus-specific IgG/IgA, limiting viral replication and immune decline and breaking the influenza–chronic disease cycle. They interrupt inflammatory amplification loops: preventing severe influenza curbs cytokine storms that worsen pathology (e.g., atherosclerotic instability and insulin resistance; see Section 3.3), maintaining metabolic balance, and reducing organ stress. Vaccines preserve mucosal barriers, lowering secondary infection risk. In COPD or patients with chronic kidney disease who are vulnerable to bacterial infections due to influenza-induced mucosal damage (see Section 3.2), they reduce respiratory epithelial injury, preserve mucociliary function, and limit colonization, cutting pneumonia/sepsis risk. In immunocompromised subgroups (e.g., chemotherapy recipients), high-dose/adjuvanted vaccines enhance antigen presentation and B-cell activation, overcoming immunosuppression to generate protective antibodies.

In summary, vaccines protect by preventing influenza and addressing interconnected vulnerabilities, restoring antiviral immunity, reducing inflammation, maintaining organ function, and blocking secondary complications, which explains their clinical benefits across chronic diseases.

## 7. Influenza Vaccination Safety and Efficacy in Immunocompromised Populations

The influenza vaccination is generally considered safe for cancer patients undergoing immunosuppressive therapies, including chemotherapy and immune checkpoint inhibitors [389]. Studies show no significant increase in adverse reaction rates among cancer patients compared to the general population [390,391,392]. Organ transplant recipients, due to immunosuppression, face elevated infection risks [393,394,395]. Influenza vaccination effectively reduces infection rates and intensive care unit (ICU) admission rates. The American Society of Organ Transplantation recommends annual influenza vaccination with priority for high-dose inactivated vaccines, which is supported by evidence of excellent tolerability in transplant patients [396,397]. Adverse reactions are typically mild, manifesting as local pain or fever (Table 5).

Inactivated vaccines are preferred for immunosuppressed individuals as they lack live viral components, eliminating the risk of viral replication. Research indicates that inactivated influenza vaccines, particularly subunit formulations, are safe for all immunosuppressed patients, regardless of their immunosuppressive therapy regimen. This includes oncology patients undergoing chemotherapy, solid organ transplant recipients on maintenance immunosuppression, and HIV-infected individuals [363,364,365].

## 8. Conclusions

Annual influenza vaccination is essential for individuals with chronic medical conditions, as it substantially reduces the risk of severe influenza-related complications, including pneumonia, myocardial infarction, and other life-threatening outcomes, while concurrently decreasing hospitalization rates and mortality. Vaccination provides significant clinical and economic benefits for patients with chronic disease by maintaining disease stability, reducing acute exacerbations, and improving quality of life. Furthermore, it decreases the incidence of influenza and its related complications, resulting in substantial healthcare cost savings.

Despite proven benefits of influenza vaccination in patients with chronic disease, critical gaps remain. Disease-specific pathophysiologies (e.g., immune dysregulation in diabetes and airway inflammation in COPD) and their impact on vaccine responses are poorly understood, limiting tailored approaches. Efficacy data are sparse for advanced heart or kidney disease, and long-term effects on chronic disease progression (e.g., atherosclerosis, COPD remodeling) are unclear. Antibody titers may not fully reflect protection, as cellular immunity or inflammation reduction may play larger roles.

Future work should (1) clarify disease-specific vaccine response barriers; (2) develop adjuvanted/high-dose subunit vaccine for immunocompromised patients; (3) study the role of vaccinations in modifying chronic disease trajectories; (4) identify novel correlates (e.g., cytokine shifts and T-cell dynamics); and (5) achieve broader or more durable protection, overcoming vaccine effectiveness variability, and improving responses in severely immunocompromised individuals. These steps will optimize benefits in this high-risk group.

## Figures and Tables

**Figure 1 vaccines-13-00936-f001:**
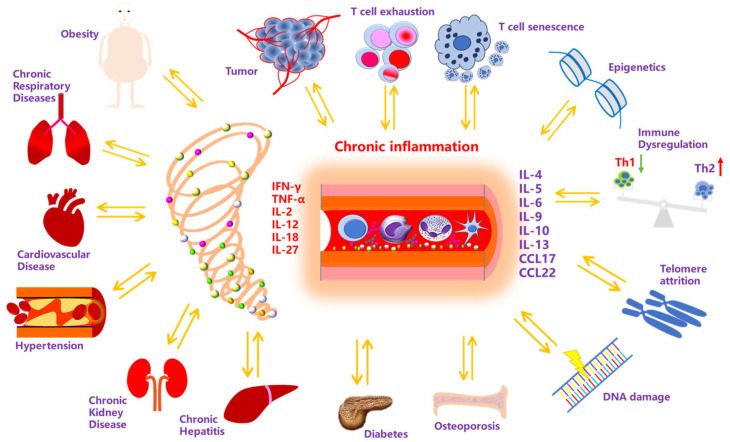
A schematic illustrating chronic inflammation as a central hub bridging metabolic, inflammatory, and immunological disorders. Chronic inflammation acts as a central mediator, bidirectionally interacting with diseases (obesity, chronic respiratory diseases, cardiovascular disease, hypertension, chronic kidney disease, chronic hepatitis, diabetes, osteoporosis, and tumors) and triggering immunological/molecular alterations (T-cell exhaustion, senescence, epigenetic changes, Th1/Th2 imbalance, telomere attrition, and DNA damage). Key inflammatory factors are categorized into Th1–associated proinflammatory cytokines (IFN–γ, TNF–α, IL–2, IL–12, IL–18, and IL–27) and Th2–regulatory subsets (IL–4, IL–5, IL–6, IL–9, IL–10, IL–13, CCL17, and CCL22), which orchestrate pathological interactions. Note: Arrows in the figure are presented in one–way/two–way forms to indicate the strength of causal relationships or feedback regulation in basic research and clinical findings, and the node size reflects its biological status.

**Figure 2 vaccines-13-00936-f002:**
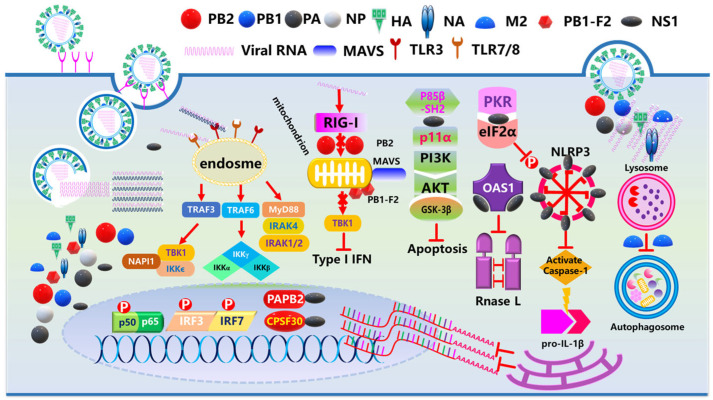
Schematic diagram of influenza virus inhibiting host cell immune responses. The influenza virus contains key proteins (PB2, PB1, PA, NP, HA, NA, M2, PB1-F2, NS1) and viral RNA; after entering host cells, viral RNA is encapsulated in endosomes and recognized by pattern recognition receptors (PRRs, including TLR3/7/8 and RIG-I), where TLRs activate NF-κB/IRF pathways via MyD88/TRIF to promote cytokines and interferons, while RIG-I interacts with MAVS, activates TBK1/IKKε, phosphorylates IRF3/IRF7, and induces type I IFN. Viral proteins such as PB1-F2 interfere with protein translation via p11α/eIF2α and modulate immune signaling via MAVS; host defenses include PKR (which inhibits protein synthesis and regulates NF-κB), OAS1 (which degrades RNA), and the NLRP3 inflammasome (which activates Caspase-1 and promotes IL-1β). The virus also induces apoptosis (regulated by Caspases) and autophagy (via autophagosome formation), and these processes may affect viral clearance or host resource utilization.

**Table 1 vaccines-13-00936-t001:** Immunological dysregulation and associated pathological mechanisms in chronic diseases.

Abnormality Type	Key Features	Associated Diseases	Molecular Mechanisms	Reference
Cellular Immunity Dysfunction	Th17/Treg imbalance; NK cell dysfunction; Reduced CD8^+^ T-cell cytotoxicity	COPD; Rheumatoid arthritis; SLE	Th17 cells secrete proinflammatory cytokines (IL-17/IL-22); Treg dysfunction leads to loss of immune tolerance; CD8^+^ T cells exhibit mitochondrial dysfunction	[59,60,61,62,63,64,65,66,67,68,69,70,71,72,73,74,75]
Humoral Immunity Dysfunction	Decreased IgG/IgA/IgM levels; Impaired antibody response	Chronic infections; Autoimmune diseases	Abnormal B cell activation; Reduced antibody production; Impaired pathogen clearance	[76,77,78,79,80,81]
Cytokine Dysregulation	Persistent overexpression of IL-6; IL-1β; and TNF-α; Proinflammatory/anti-inflammatory imbalance	Cardiovascular diseases; Diabetes; SLE	NF-κB pathway activation → excessive cytokine release → tissue damage and chronic inflammation	[82,83,84,85,86,87,88,89,90]
Autoimmune Reactions	Elevated autoantibodies (e.g., anti-nuclear antibodies); Activation of autoreactive T/B cells	Rheumatoid arthritis; Type 1 diabetes	Breakdown of immune tolerance; Attack on self-antigens	[91,92,93,94,95,96,97,98,99]
Signaling Pathway Abnormalities	Overactivation of TLR/NF-κB pathways; Dysregulation of JAK-STAT pathways	Autoimmune diseases; Chronic inflammation	TLR recognizes pathogens → NF-κB → proinflammatory cytokine release- JAK-STAT mediates abnormal cytokine signaling	[100,101,102,103,104,105,106,107,108,109,110,111,112]
Epigenetic Mechanisms	Abnormal DNA methylation/histone modification; Dysregulated immune-related gene expression	Chronic fatigue syndrome; Cancer	HDAC activity changes; gene silencing or activation	[113,114,115,116,117,118,119]
Microenvironmental Influences	Disrupted cytokine/chemokine microenvironment; Tissue fibrosis or remodeling	COPD; Pulmonary fibrosis	Inflammatory mediators (e.g., TGF-β) promote abnormal tissue repair, organ dysfunction	[120,121,122,123,124,125,126]

Note: The arrows in this table denote a sequential causal relationship within the biological process.

**Table 2 vaccines-13-00936-t002:** Pathological mechanisms and clinical consequences of influenza infection in patients with chronic disease.

Mechanism Category	Pathological Mechanism	Associated Chronic Diseases	Clinical Symptoms and Hazards	Reference
Inflammatory Response and Cytokine Storm	Influenza virus activates the immune system, releasing proinflammatory cytokines (IL-6, TNF-α) that trigger systemic inflammatory response syndrome (SIRS).	Cardiovascular diseases; COPD/asthma; autoimmune diseases	Cardiovascular: Plaque rupture and myocardial fibrosis; COPD/asthma: Airway spasm and increased mucus production; Autoimmune diseases: Activation of autoimmune responses.	[168,169,209,239,240]
Immune System Dysfunction and Secondary Infections	Viral suppression of interferon signaling leads to T-cell exhaustion and secondary bacterial infections (e.g., Streptococcus pneumoniae).	Diabetes; chronic kidney/liver diseases	Diabetes: Increased risk of severe pneumonia; Kidney/liver diseases: Increased sepsis risk and accelerated organ failure.	[241,242]
Metabolic and Homeostatic Dysregulation	Infection activates the HPA axis, leading to insulin resistance and increased catabolism.	Diabetes; heart failure	Diabetes: Stress-induced hyperglycemia and diabetic ketoacidosis; Heart failure: Sympathetic activation → increased cardiac workload.	[243,244,245,246,247,248,249,250,251]
Organ Compensatory Insufficiency	Patients with chronic disease have reduced organ reserve capacity, with infection-induced metabolic demands exceeding compensatory ability.	Chronic kidney disease; chronic liver disease	Kidney disease: Dehydration/sepsis and acute kidney injury (AKI); Liver disease: Hepatocyte necrosis and hepatic encephalopathy/coagulopathy	[252,253,254,255,256]
Coagulation Abnormalities	The virus activates endothelial cells and platelets, inducing a hypercoagulable state (tissue factor release).	Cardiovascular diseases; malignant tumors	Cardiovascular: Increased risk of deep vein thrombosis (DVT) and pulmonary embolism (PE); Tumors: Superimposed thrombotic risk	[257,258,259,260]
Drug Interactions	Immunosuppressants (e.g., glucocorticoids) inhibit antiviral immunity; β-blockers mask infection symptoms.	Patients using immunosuppressants/ cardiovascular drugs	Prolonged viral replication time; Delayed infection diagnosis.	[261,262,263]

**Table 3 vaccines-13-00936-t003:** Indicators for patients with chronic disease during influenza infection.

Health Indicator	Disease Category	Statistical Value	Risk Comparison (vs. Healthy Population)	Reference
Hospitalization Rate	Chronic Respiratory Diseases	3–5× higher risk	Accounts for 21.4% of total hospitalizations	[299]
	Cardiovascular Diseases	2.8x higher rate	Contributes to 18.6% of hospitalization causes	[23,300]
Severity Rate	Diabetes	40% higher severity conversion rate	Accounts for 63% of influenza severe cases	[301,302]
	Elderly Chronic Patients (≥65 years)	12.7% severity rate	Contributes to 71% of ICU cases	[296,303]
Mortality Rate	Overall Chronic Patients	82% of influenza deaths	February 2025 mortality rate: 0.01%	[304]
	Cardiovascular and Cerebrovascular Diseases	278.6 per 100,000	1.8x higher death risk	[305,306]
	Cancer Patients	320 per 100,000	3.2x higher mortality	[307,308]
Special Risk Factors	Hypertension (Awareness Rate: 45.72%)	Significantly increases complication risk	/	[309,310,311,312,313]
	Chronic Patients with Smoking History	55% higher severity rate	/	[314,315,316,317,318,319]

**Table 4 vaccines-13-00936-t004:** Vaccination efficacy and clinical benefits across high-risk populations *.

Population Group		Hospitalization Rate Reduction (vs. Unvaccinated in the Same Group)	Mortality Rate Reduction (vs. Unvaccinated in the Same Group)	Vaccine Types	Mechanism of Vaccine Protection	References
Elderly (≥65 years)		43–52%	38–56%	High-dose/adjuvanted vaccines	Enhanced immunogenicity in immunosenescent populations	[338]
Children (6 months-5 years)		40–70%	65–75%	Standard pediatric formulations	Immature immune system priming	[339]
Patients with Chronic Disease	Cardiovascular Disease	30–56%	18–34%	Standard formulations	Reduction in cardiovascular stress during infection	[340,341,342,343]
	COPD Patients	32–52%	50–70%	Standard formulations	Alleviation of respiratory impairment during infection	[344,345,346,347,348,349,350,351]
	Diabetes (Type 1/2)	23–58%	16–46%	Standard formulations	Glycemic stabilization during infection	[352,353,354,355]
	Obesity	20–40%	10–25%	Standard formulations	Mitigation of metabolic stress during infection	[205,356,357,358]
	Chronic Kidney Disease	11–30%	15–30%	Standard formulations	Mitigation of renal burden and dialysis dependence	[359,360,361,362]
	HIV/AIDS	20–40%	10–25%	High-dose vaccines	Immune reconstitution effects	[363,364,365,366,367]
	Post-Organ Transplant	20–40%	10–40%	Standard formulations	Immunosuppression modulation	[368,369,370,371]
	Chronic Liver Disease	27–50%	10–20%	Standard formulations	Hepatic protective effects	[372,373,374,375]
	Hematologic Diseases	20–40%	10–25%	Standard formulations	Reduction in infection-induced disease exacerbations	[376,377]
	Systemic Lupus Erythematosus	18–52%	40–59%	Standard formulations	Autoimmune modulation effects	[378,379,380,381,382]

* All data represent comparisons between vaccinated and unvaccinated patients within the same high-risk/chronic disease group.

**Table 5 vaccines-13-00936-t005:** Influenza vaccine seroconversion rates across different populations.

Population Group		H1N1 Seroconversion Rate	H3N2 Seroconversion Rate	B-Type Influenza Seroconversion Rate	Vaccine Type	References
Healthy Adults		40–60%	30–50%	35–55%	Standard inactivated vaccine	[398,399,400,401,402]
Elderly (≥65 years)		40–50%	30–40%	25–35%	High-dose/adjuvanted vaccine	[403,404,405,406,407]
Children (6 months-5 years)		40–70%	65–75%	50–70%	Pediatric formulation	[408,409,410,411,412,413]
Patients with Chronic Disease	Cardiovascular Disease	30–50%	20–40%	25–45%	Standard vaccine	[414,415,416,417,418,419,420]
	COPD	35–55%	25–45%	30–50%	Standard vaccine	[345,421,422,423,424]
	Diabetes (Type 1/2)	24–58%	18–42%	20–40%	High-dose vaccine	[354,425,426,427,428]
	Obesity	30–50%	20–40%	25–45%	Standard vaccine	[356,429,430,431]
	Chronic Kidney Disease	50–60%	40–50%	30–40%	Standard vaccine	[359,361,432,433,434]
	HIV/AIDS	40–60%	30–50%	30–50%	High-dose vaccine	[435]
	Post-Organ Transplant	40–60%	30–50%	20–40%	Standard vaccine	[436]
	Chronic Liver Disease	35–55%	25–45%	30–50%	Standard vaccine	[372,373,437]
	Hematologic Diseases	30–50%	20–40%	25–45%	Standard vaccine	[438,439]
	Systemic Lupus Erythematosus	35–55%	25–45%	30–50%	Standard vaccine	[382,440,441,442,443,444,445]
	Malignant Tumors	30–50%	20–40%	10–30%	Standard/high-dose vaccine	[446,447,448]

## Data Availability

The data are contained within the article. The raw data supporting the conclusions of this article will be made available by the authors on request.

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
