# Peer review of "Chronic Diseases and Influenza Vaccines"

_vaccines, 2025, doi:10.3390/vaccines13090936_

Round 1

Reviewer 1 Report

Comments and Suggestions for Authors

The review article by Dr. Lian et al provides a comprehensive overview of how chronic diseases influence influenza-related effects. It highlights how influenza infection, in turn, worsens chronic conditions and role of vaccination in mitigating these impacts. The paper is well-structured, covering the core areas of such interaction of influenza and chronic diseases. However, I have following suggestions to consider improving the article.

  • As Section 3.1 describes various immune dysregulations in chronic diseases followed by influenza's immunopathological injuries (as in 3.2), it will be great to provide an integrated discussion of how these two sets of mechanisms interact each other.
  • Please remove some redundancies, e.g. Section 3.3 discusses how influenza exacerbates specific chronic diseases and describes general mechanisms that are already discussed in Section 3.2 as general effects of influenza infection. Please rectify this.
  • Section 6 discusses efficacy of influenza vaccines in reducing mortality in chronic disease patients. However, the explanations for how these vaccines achieve protection, particularly in the context of complex chronic diseases, will be good to add on discussion.
  • Adding a section on limitations of current knowledge or areas for future research will improve the article,e.g., a dedicated segment discussing current challenges in influenza vaccine development (e.g., achieving broader or more durable protection, overcoming vaccine effectiveness variability, improving responses in severely immunocompromised individuals) should be added.
  • Figure legends are extremely long, for eye-catch and concentration, its good to reduce them by adding rest details in text.

Author Response

1. As Section 3.1 describes various immune dysregulations in chronic diseases followed by influenza's immunopathological injuries (as in 3.2), it will be great to provide an integrated discussion of how these two sets of mechanisms interact each other.

Thank you very much for your valuable comments. We described immune dysregulation induced by chronic diseases in Section 3.1, and direct immunopathological injury and hypersensitive reactions induced by influenza in Section 3.2. We discussed how influenza infection exacerbates chronic diseases in Section 3.3, where we also added relevant information on how these two sets of mechanisms interact with each other. The added discussion is as follows:

Immune dysregulation associated with chronic diseases and immunopathological injuries caused by influenza interact dynamically, forming a vicious cycle. Chronic diseases (see Section 3.1) create a "primed" inflammatory microenvironment characterized by elevated proinflammatory cytokines and impaired mucosal barriers, which amplifies the immunopathological effects of influenza. This pre-existing state hypersensitizes the immune system to viral triggers, exacerbating cytokine storms and tissue damage during influenza infection.

Conversely, influenza disrupts immune homeostasis in these patients (see Section 3.2): the virus' suppression of adaptive immunity (such as targeted infection of activated lymphocytes) exacerbates pre-existing humoral immune deficiencies, weakening antiviral defense capabilities and the control of underlying diseases. Additionally, influenza-induced epithelial barrier disruption and secondary infections exploit the compromised mucosal immunity in chronic respiratory diseases like COPD, accelerating airway remodeling and fibrosis.

This reciprocal interaction increases the risk of severe influenza complications and exacerbation of underlying chronic diseases in patients with chronic illnesses, highlighting the necessity of optimizing influenza vaccination strategies to break this cycle.

2. Please remove some redundancies, e.g. Section 3.3 discusses how influenza exacerbates specific chronic diseases and describes general mechanisms that are already discussed in Section 3.2 as general effects of influenza infection. Please rectify this.

According to your comments, we remove some redundancies in Section 3.3.

3. Section 6 discusses efficacy of influenza vaccines in reducing mortality in chronic disease patients. However, the explanations for how these vaccines achieve protection, particularly in the context of complex chronic diseases, will be good to add on discussion.

Thank you for your valuable comment regarding Section 6. We fully agree that elaborating on how influenza vaccines achieve protection, especially in the context of complex chronic diseases, would enhance the discussion. We therefore add relevant explanations in this section to address this point, and we appreciate your insightful suggestion which helps improve the manuscript. According to your suggestion, we have already added the explanations in the main text for how these vaccines achieve protection, particularly in the context of complex chronic diseases and we have also listed the added content here as follows:

Influenza vaccines reduce hospitalization and mortality in chronic disease patients by targeting their unique vulnerabilities, countering immune dysregulation, metabolic imbalances, and organ-specific weaknesses to mitigate influenza-induced exacerbations. Vaccines address inherent immune deficits: in diabetics or cardiac patients with impaired CD8⁺ T-cell activity and excess pro-inflammatory cytokines (see Section 3.1), they restore CD8⁺ T-cell-mediated clearance and induce virus-specific IgG/IgA, limiting viral replication and immune decline, breaking the influenza-chronic disease cycle. They interrupt inflammatory amplification loops: preventing severe influenza curbs cytokine storms that worsen pathology (e.g., atherosclerotic instability, insulin resistance; Section 3.3), maintaining metabolic balance and reducing organ stress. Vaccines preserve mucosal barriers, lowering secondary infection risk. In COPD or chronic kidney disease patients-vulnerable to bacterial infections from influenza-induced mucosal damage (see Section 3.2) they reduce respiratory epithelial injury, preserve mucociliary function, and limit colonization, cutting pneumonia/sepsis risk.  In immunocompromised subgroups (e.g., chemotherapy recipients), high-dose/adjuvanted vaccines enhance antigen presentation and B-cell activation, overcoming immunosuppression to generate protective antibodies. 

In summary, vaccines protect by preventing influenza and addressing interconnected vulnerabilities-restoring antiviral immunity, reducing inflammation, maintaining organ function, and blocking secondary complications-explaining clinical benefits across chronic diseases.

4. Adding a section on limitations of current knowledge or areas for future research will improve the article, e.g., a dedicated segment discussing current challenges in influenza vaccine development (e.g., achieving broader or more durable protection, overcoming vaccine effectiveness variability, improving responses in severely immunocompromised individuals) should be added.

Thank you for your constructive suggestion regarding the inclusion of a section on limitations of current knowledge and areas for future research. We fully agree that adding such a dedicated segment will significantly enhance the comprehensiveness of the article. In particular, we recognize the value of discussing current challenges in influenza vaccine development, including achieving broader or more durable protection, addressing variability in vaccine effectiveness, and improving responses in severely immunocompromised individuals. We incorporated this section to highlight these critical aspects, and we appreciate your insightful input, which will help strengthen the manuscript. According to your suggestion, we have already added the explanations in the main text for how these vaccines achieve protection, particularly in the context of complex chronic diseases and we have also listed the added content here as follows:

Despite proven benefits of influenza vaccination in chronic disease patients, critical gaps remain. Disease-specific pathophysiologies (e.g., immune dysregulation in diabetes, airway inflammation in COPD) and their impact on vaccine responses are poorly understood, limiting tailored approaches. Efficacy data are sparse for advanced heart or kidney disease, and long-term effects on chronic disease progression (e.g., atherosclerosis, COPD remodeling) are unclear. Antibody titers may not fully reflect protection, as cellular immunity or inflammation reduction may play larger roles. 

Future work should: (1) Clarify disease-specific vaccine response barriers; (2) Develop adjuvanted/high-dose vaccines for immunocompromised patients; (3) Study role of vaccinations in modifying chronic disease trajectories; (4) Identify novel correlates (e.g., cytokine shifts, T-cell dynamics); (5) Achieving broader or more durable protection, overcoming vaccine effectiveness variability, improving responses in severely immunocompromised individuals. These steps will optimize benefits in this high-risk group.

5. Figure legends are extremely long, for eye-catch and concentration, its good to reduce them by adding rest details in text.

Thank you very much for your valuable suggestion regarding the figure legends. Following your advice, we have carefully revised the figure legends by streamlining them to highlight only the most essential details necessary for immediate understanding of the figure.

Reviewer 2 Report

Comments and Suggestions for Authors

The manuscript “Chronic Diseases and Influenza Vaccines”, submitted to the journal Vaccines , is devoted to the reviewing the chronic conditions  that exacerbate influenza-related effects through immune dysfunction and metabolic imbalances, and how influenza infection worsens chronic diseases by triggering inflammation, suppressing immunity, and causing secondary infections that lead to respiratory complications, cardiac complications, and blood sugar disturbances.  Additionally, it discusses the proven benefits of vaccination. The theme is of great importance for the improvement the effectiveness of vaccination against influenza infection.

I have some comments:

  1. Lines 70-71: “However, the influenza vaccination rate among chronic disease patients in China is far lower than that in developed countries [27]”.

Comment:

It is advisable to provide specific data.

  1. Lines 92-93:

“The prevalence rate of diabetes is 20.1%, with more than 100 million patients”

Comment:

China's population is over 1 billion people 20.1% of the population is much more than 100 million people.

  1. References must be cited in order. In the manuscript, source 64 is followed by source 78 (in Table 1). In Tables 1 -5, the number of references must be changed in the References section.
  2. Lines 196-198:

“Iwasaki & Medzhitov and Herold et al. proposed the central role of PRRs in innate immunity, summarizing three key signaling pathways:”

Comment:

Probably needs to add "in viral RNA recognition"

  1. It is desirable to illustrate the discussed pathways of recognition of intracellular viral RNA with a figure.
  2. Reviews must include pictures. Illustrate the main points with pictures.
  3. Format the list of references in accordance with the requirements.

Author Response

1. Lines 70-71: However, the influenza vaccination rate among chronic disease patients in China is far lower than that in developed countries [27]. Comment: It is advisable to provide specific data.

Thank you very much for your feedback; we have added the specific data in the main text and have also listed the added content here as follows:

Among patients with chronic diseases, the influenza vaccination rate is approximately 11.5% in China, whereas it ranges from 50% to 60% in developed countries.

2. Lines 92-93:The prevalence rate of diabetes is 20.1%, with more than 100 million patientsComment: China's population is over 1 billion people 20.1% of the population is much more than 100 million people.

Thank you for your careful review and this important observation regarding the data on diabetes prevalence in lines 92-93. We greatly appreciate your attention to this detail.

You are completely correct. Based on China's total population, a prevalence rate of 20.1% would correspond to over 100 million patients. This discrepancy stems from an oversight in our initial description. We revised this section to clearly specify the population range. In fact, according to the relevant reference, the number of diabetes patients is over 2.3 billion. We have therefore revised "100 million" to "2.3 billion" in the main text to ensure the description is more accurate.

3. References must be cited in order. In the manuscript, source 64 is followed by source 78 (in Table 1). In Tables 1 -5, the number of references must be changed in the References section.

We made the necessary revision according to your suggestion.

4, Lines 196-198:Iwasaki & Medzhitov and Herold et al. proposed the central role of PRRs in innate immunity, summarizing three key signaling pathways:Comment: Probably needs to add "in viral RNA recognition"It is desirable to illustrate the discussed pathways of recognition of intracellular viral RNA with a figure. Reviews must include pictures. Illustrate the main points with pictures.

Thank you sincerely for your constructive feedback regarding figure illustration and reference formatting, which we fully recognize as critical for clarity and compliance.

The part described Figure 1 of the article has been marked in accordance with your suggestions. In addition, Figure 1 in the manuscript already details the pathways of intracellular viral RNA recognition (e.g., PRR-mediated signaling via TLR3/7/8 and RIG-I, and their interactions with viral/host factors), with complementary discussions in the text (see Section 3.2).

5. Format the list of references in accordance with the requirements.

Thank you sincerely for your careful review. We have carefully reviewed and formatted the reference list to strictly adhere to the journal’s requirements, ensuring both consistency and full compliance with the specified guidelines.

Reviewer 3 Report

Comments and Suggestions for Authors

This review titled “chronic disease and influenza vaccine” summarizes how significant patients with chronic disease should receive the influenza vaccine. This reports described the global trend of chronic disease in China and Global, immune dysregulation induced by chronic disease, how influenza virus induced immunopathological injury, influenza infection exacerbate chronic disease and pathological mechanism and consequence after influenza infection in chronic disease, the epidemiology of influenza virus infection in patients with chronic disease, and efficacy of influenza vaccine in patients with chronic disease. While the content is well-written, some rearrangements are necessary to improve the flow of ideas. 

Below are additional comments:

1.     In the introduction, while the author highlights the significance of chronic diseases in the public health burden, the manuscript lacks a clear explanation of the importance of the seasonal influenza vaccine in preventing or reducing disease severity and mortality among chronic disease patients. Please briefly discuss the role of influenza vaccination in mitigating complications of chronic diseases.

2.     To enhance the manuscript's clarity, the authors should consider a comprehensive restructuring of the content, particularly in lines 233-404. The recommended revision involves creating a new section titled "Influenza Virus and Immunopathological Interactions" that explains viral factors increasing pathogenicity, viral protein-mediated immune suppression mechanisms, and detailed immunological interactions. This section should be followed by a clearly delineated exploration of pathophysiological consequences, organized into three distinct subsections: (1) respiratory system impact, detailing chronic respiratory condition exacerbations and airway inflammation mechanisms; (2) cardiovascular system implications, addressing inflammatory responses and cardiovascular burden; and (3) metabolic disruptions, explaining glycemic dysregulation, insulin resistance, and influenza-induced metabolic stress. A subsequent section on Immunological Complications should comprehensively cover hypersensitivity reactions, cytokine storm, and secondary infection dynamics.

3.     The authors mention immune dysregulation and associated chronic diseases in lines 106-177. To enhance clarity, the cited references in the text should be aligned with the references in Table 1.

4.     Ensure that the table heading appears on the subsequent table on the next page of the manuscript to maintain the clarity and readability of the table's content. 

5.     In lines 431-433, the statement "These findings underscore the critical importance of: (1) increasing influenza vaccination…” is not consistent with the previous context that described hospitalization, severity, and mortality in chronic disease patients versus the general population. The summary in lines 431-433 should explicitly reflect the context, demonstrating that influenza vaccination reduces hospitalization, disease severity, and mortality in patients with chronic diseases compared to those who did not receive the influenza vaccine. Please clarify.

6.     In lines 460-470 and Table 4, the data on hospitalization and mortality rate reductions lack clarity about the comparison population. Please clearly specify whether the comparison is made (i) between vaccinated patients with chronic disease and unvaccinated patients with chronic disease, or (ii) between vaccinated patients with chronic disease and the general population.

Author Response

This review titled “chronic disease and influenza vaccine” summarizes how significant patients with chronic disease should receive the influenza vaccine. This reports described the global trend of chronic disease in China and Global, immune dysregulation induced by chronic disease, how influenza virus induced immunopathological injury, influenza infection exacerbate chronic disease and pathological mechanism and consequence after influenza infection in chronic disease, the epidemiology of influenza virus infection in patients with chronic disease, and efficacy of influenza vaccine in patients with chronic disease. While the content is well-written, some rearrangements are necessary to improve the flow of ideas.

Thank you sincerely for your thoughtful and detailed feedback on our review. We greatly appreciate your recognition of the content coverage. Your suggestion regarding rearranging sections to enhance the flow of ideas is valuable, and we fully agree that improving the logical progression will strengthen the manuscript. We carefully revisit the structure of the review, ensuring that each section transitions smoothly into the next.  We are grateful for your guidance, which will undoubtedly help elevate the quality of our work. Thank you again for your time and expertise.

Below are additional comments:

1. In the introduction, while the author highlights the significance of chronic diseases in the public health burden, the manuscript lacks a clear explanation of the importance of the seasonal influenza vaccine in preventing or reducing disease severity and mortality among chronic disease patients. Please briefly discuss the role of influenza vaccination in mitigating complications of chronic diseases.

Thank you very much for your comments, we added the related information in introduction section and discuss the role of influenza vaccination in mitigating complications of chronic diseases in Section 6. Reviewer 1 also raised similar question,

the added discussion is as follows:

Influenza vaccines reduce hospitalization and mortality in chronic disease patients by targeting their unique vulnerabilities, countering immune dysregulation, metabolic imbalances, and organ-specific weaknesses to mitigate influenza-induced exacerbations. Vaccines address inherent immune deficits: in diabetics or cardiac patients with impaired CD8⁺ T-cell activity and excess pro-inflammatory cytokines (see Section 3.1), they restore CD8⁺ T-cell-mediated clearance and induce virus-specific IgG/IgA, limiting viral replication and immune decline, breaking the influenza-chronic disease cycle. They interrupt inflammatory amplification loops: preventing severe influenza curbs cytokine storms that worsen pathology (e.g., atherosclerotic instability, insulin resistance; Section 3.3), maintaining metabolic balance and reducing organ stress. Vaccines preserve mucosal barriers, lowering secondary infection risk. In COPD or chronic kidney disease patients-vulnerable to bacterial infections from influenza-induced mucosal damage (see Section 3.2) they reduce respiratory epithelial injury, preserve mucociliary function, and limit colonization, cutting pneumonia/sepsis risk.  In immunocompromised subgroups (e.g., chemotherapy recipients), high-dose/adjuvanted vaccines enhance antigen presentation and B-cell activation, overcoming immunosuppression to generate protective antibodies. 

In summary, vaccines protect by preventing influenza and addressing interconnected vulnerabilities-restoring antiviral immunity, reducing inflammation, maintaining organ function, and blocking secondary complications-explaining clinical benefits across chronic diseases.

2. To enhance the manuscript's clarity, the authors should consider a comprehensive restructuring of the content, particularly in lines 233-404. The recommended revision involves creating a new section titled "Influenza Virus and Immunopathological Interactions" that explains viral factors increasing pathogenicity, viral protein-mediated immune suppression mechanisms, and detailed immunological interactions. This section should be followed by a clearly delineated exploration of pathophysiological consequences, organized into three distinct subsections: (1) respiratory system impact, detailing chronic respiratory condition exacerbations and airway inflammation mechanisms; (2) cardiovascular system implications, addressing inflammatory responses and cardiovascular burden; and (3) metabolic disruptions, explaining glycemic dysregulation, insulin resistance, and influenza-induced metabolic stress. A subsequent section on Immunological Complications should comprehensively cover hypersensitivity reactions, cytokine storm, and secondary infection dynamics.

 Thank you very much for your valuable and detailed suggestions on restructuring the manuscript to enhance its clarity. We fully agree with your proposed framework.

We clearly delineated the exploration of pathophysiological consequences into three distinct subsections in Section 3.2: (A) respiratory system impact, with a focus on chronic respiratory condition exacerbations and airway inflammation mechanisms; (B) cardiovascular system implications, addressing inflammatory responses and cardiovascular burden; and (C) metabolic disruptions, explaining glycemic dysregulation, insulin resistance, and influenza-induced metabolic stress.

We described immune dysregulation induced by chronic diseases in Section 3.1, and direct immunopathological injury and hypersensitive reactions induced by influenza in Section 3.2. We discussed how influenza infection exacerbates chronic diseases in Section 3.3. According to your suggestion, we added a paragraph on how these two sets of mechanisms interact with each other(Influenza Virus and Immunopathological Interactions) which is also raised by reviewer 1.

The added discussion is as follows:

Immune dysregulation associated with chronic diseases and immunopathological injuries caused by influenza interact dynamically, forming a vicious cycle. Chronic diseases (see Section 3.1) create a "primed" inflammatory microenvironment characterized by elevated proinflammatory cytokines and impaired mucosal barriers, which amplifies the immunopathological effects of influenza. This pre-existing state hypersensitizes the immune system to viral triggers, exacerbating cytokine storms and tissue damage during influenza infection.

Conversely, influenza disrupts immune homeostasis in these patients (see Section 3.2): the virus' suppression of adaptive immunity (such as targeted infection of activated lymphocytes) exacerbates pre-existing humoral immune deficiencies, weakening antiviral defense capabilities and the control of underlying diseases. Additionally, influenza-induced epithelial barrier disruption and secondary infections exploit the compromised mucosal immunity in chronic respiratory diseases like COPD, accelerating airway remodeling and fibrosis.

This reciprocal interaction increases the risk of severe influenza complications and exacerbation of underlying chronic diseases in patients with chronic illnesses, highlighting the necessity of optimizing influenza vaccination strategies to break this cycle.

3. The authors mention immune dysregulation and associated chronic diseases in lines 106-177. To enhance clarity, the cited references in the text should be aligned with the references in Table 1.

Thank you for pointing out this important detail regarding the alignment of cited references in the text (lines 106-177) with those in Table 1. We fully recognize the significance of this for enhancing clarity and ensuring consistency in the manuscript.

We reviewed both the text discussions on immune dysregulation and associated chronic diseases, as well as the references listed in Table 1, to ensure complete alignment. Thank you again for your valuable feedback.

4. Ensure that the table heading appears on the subsequent table on the next page of the manuscript to maintain the clarity and readability of the table's content.

Thank you for your valuable suggestion regarding the table heading. We appreciate your attention to this detail. We will ask magazine layout editor carefully adjust the manuscript layout to ensure that the table heading is positioned together with the corresponding table on the next page, eliminating any separation between them.

5. In lines 431-433, the statement "These findings underscore the critical importance of: (1) increasing influenza vaccination…” is not consistent with the previous context that described hospitalization, severity, and mortality in chronic disease patients versus the general population. The summary in lines 431-433 should explicitly reflect the context, demonstrating that influenza vaccination reduces hospitalization, disease severity, and mortality in patients with chronic diseases compared to those who did not receive the influenza vaccine. Please clarify.

Thank you for your insightful comment, which helps enhance the consistency and targeting of our conclusion. We fully agree that the original statement failed to explicitly link to the preceding context-where we compared influenza-related hospitalization, disease severity, and mortality between patients with chronic diseases and the general population-leading to insufficient alignment between the summary and the core content.

To address this, we have revised the summary in lines 431-433 to specifically focus on "patients with chronic diseases" (the key population emphasized in the previous context) and add a clear comparison with "unvaccinated chronic disease patients," so as to directly reflect the contextual logic. The revised statement is as follows:

"These findings underscore the critical importance of: (1) increasing influenza vaccination coverage in patients with chronic diseases (a key high-risk group) to reduce their influenza-related hospitalization, disease severity, and mortality compared to their unvaccinated counterparts; and (2) strengthening influenza surveillance systems and implementing early intervention strategies to further improve clinical outcomes in this vulnerable population."

This revision explicitly ties the vaccination recommendations to the prior analysis of chronic disease patients’ influenza burden, ensuring the summary accurately echoes the context and clarifies the specific benefits of influenza vaccination for this group.

6. In lines 460-470 and Table 4, the data on hospitalization and mortality rate reductions lack clarity about the comparison population. Please clearly specify whether the comparison is made (i) between vaccinated patients with chronic disease and unvaccinated patients with chronic disease, or (ii) between vaccinated patients with chronic disease and the general population.

Thank you for identifying the ambiguity in the comparison population. We apologize for this oversight and have clarified the comparison framework as follows:

(i)Rationale for the Comparison

All data in lines 460–470 and Table 4 refer to comparison (i): vaccinated patients with a specific chronic disease vs. unvaccinated patients with the same chronic disease. This aligns with the table’s title (“Vaccination Efficacy and Clinical Benefits Across High-Risk Populations”) and the subpopulation structure (e.g., “Cardiovascular Disease,” “Diabetes,” etc.), where efficacy is evaluated within each high-risk group (not against the general population).

(ii)Revisions to the Text (Lines 460–470)

We have added explicit comparisons to the text:

Multiple studies have demonstrated that influenza vaccination significantly reduces hospitalization and mortality risks associated with influenza and related complications in patients with chronic diseases, compared with unvaccinated patients with the same chronic diseases [263]. A meta-analysis showed that influenza vaccination reduces all-cause mortality by 46% and influenza/pneumonia-related hospitalization by 45% in elderly patients with diabetes, compared with unvaccinated elderly patients with diabetes [264]. For individuals with cardiovascular diseases, influenza vaccines exert protective effects by lowering the risk of adverse cardiovascular events, with a 34% reduction in cardiovascular event risk observed among vaccinated patients, compared with unvaccinated individuals with cardiovascular diseases. Additionally, the increased incidence of acute myocardial infarction and mortality during influenza seasons highlights the strong association between cardiovascular diseases and influenza infection [265] (Table 4).

To ensure clarity, we have revised the column headers and added an explanatory footnote to Table 4: the header “Hospitalization Rate Reduction” has been updated to “Hospitalization Rate Reduction (vs. unvaccinated in the same group)”, and “Mortality Rate Reduction” to “Mortality Rate Reduction (vs. unvaccinated in the same group)”. Additionally, a footnote (denoted as a) has been included below the table, specifying: “All data represent comparisons between vaccinated and unvaccinated patients within the same high-risk or chronic disease group.”